# SpineHRformer: A Transformer-Based Deep Learning Model for Automatic Spine Deformity Assessment with Prospective Validation

**DOI:** 10.3390/bioengineering10111333

**Published:** 2023-11-20

**Authors:** Moxin Zhao, Nan Meng, Jason Pui Yin Cheung, Chenxi Yu, Pengyu Lu, Teng Zhang

**Affiliations:** Department of Orthopaedics and Traumatology, The University of Hong Kong, Hong Kong; moxin@connect.hku.hk (M.Z.); nanmeng@hku.hk (N.M.); cheungjp@hku.hk (J.P.Y.C.); ychxhku@hku.hk (C.Y.); pylu@hku.hk (P.L.)

**Keywords:** HRNet, transformer, Cobb angle automatic measurement, endplate detection, end vertebrae detection

## Abstract

The Cobb angle (CA) serves as the principal method for assessing spinal deformity, but manual measurements of the CA are time-consuming and susceptible to inter- and intra-observer variability. While learning-based methods, such as SpineHRNet+, have demonstrated potential in automating CA measurement, their accuracy can be influenced by the severity of spinal deformity, image quality, relative position of rib and vertebrae, etc. Our aim is to create a reliable learning-based approach that provides consistent and highly accurate measurements of the CA from posteroanterior (PA) X-rays, surpassing the state-of-the-art method. To accomplish this, we introduce SpineHRformer, which identifies anatomical landmarks, including the vertices of endplates from the 7th cervical vertebra (C7) to the 5th lumbar vertebra (L5) and the end vertebrae with different output heads, enabling the calculation of CAs. Within our SpineHRformer, a backbone HRNet first extracts multi-scale features from the input X-ray, while transformer blocks extract local and global features from the HRNet outputs. Subsequently, an output head to generate heatmaps of the endplate landmarks or end vertebra landmarks facilitates the computation of CAs. We used a dataset of 1934 PA X-rays with diverse degrees of spinal deformity and image quality, following an 8:2 ratio to train and test the model. The experimental results indicate that SpineHRformer outperforms SpineHRNet+ in landmark detection (Mean Euclidean Distance: 2.47 pixels vs. 2.74 pixels), CA prediction (Pearson correlation coefficient: 0.86 vs. 0.83), and severity grading (sensitivity: normal-mild; 0.93 vs. 0.74, moderate; 0.74 vs. 0.77, severe; 0.74 vs. 0.7). Our approach demonstrates greater robustness and accuracy compared to SpineHRNet+, offering substantial potential for improving the efficiency and reliability of CA measurements in clinical settings.

## 1. Introduction

Adolescent idiopathic scoliosis (AIS) is the most prevalent type of spinal deformity, affecting 1–3% of children aged 10–16 years, with a higher incidence among girls [1,2]. AIS is characterized by lateral curvature of the spine [3] and is believed to arise from a multifaceted interplay of genetic and environmental factors [4,5]. Due to its rapid progression during puberty [2,6], timely detection, intervention, and regular follow-ups are crucial in mitigating its progression and minimizing potential complications associated with the disorder [1,7,8].

The Cobb angle (CA) [9] is an essential reference for healthcare professionals in the assessment and management of spinal deformity, enabling the identification of the most appropriate treatment strategies for individual scoliosis patients. To derive the CA of one spine curve, the expert first needs to identify the two end vertebrae which exhibit the highest degree of tilt from the horizontal axis. Then, the CA can be obtained by measuring the angle formed between the line across the upper endplate of the upper end vertebra and the line across the lower endplate of the lower end vertebra. However, the process requires multiple steps and is highly dependent on the observer’s skills and experience, making it time-consuming and inconsistent between inter- and intra-observers [10]. Automated methods for accurately measuring the CAs are thus imperative.

Existing CA auto-measurement methods can be categorized into segmentation-based methods [11,12,13,14,15,16,17,18,19], regression-based methods [18,20,21,22,23], and heatmap-based methods [24,25]. In most of the segmentation-based methods, each vertebra is segmented, then the upper and lower fitting lines of the area are extracted to obtain the slopes of the upper and lower endplates. According to the slopes of the endplates, the end vertebrae can be determined and then the CA can be measured. To realize the vertebra segmentation, both traditional image processing algorithms [13,14] and learning-based algorithms have been used [11,15,16,17,18,26]. However, the accuracy of such methods is questionable, since the process of obtaining CAs accumulates the error during segmentation and endplate slope calculation [23].

Regression-based techniques aim to directly predict the coordinates of endplate vertices by employing deep learning models. These models utilize multiple layers to incrementally extract higher-level features from the input data. The CA is either obtained by another regression model with the coordinates as its input or obtained by the slopes of endplates being calculated with the coordinates [18,20,21,22,23]. The determination of the CA accuracy highly relies on the precision of endplate landmark prediction. To improve the model performance, some researchers reduce the influence of image outliers by splitting the image into patches and predicting the landmarks in the patches. Wu et al. [20] detected the vertebral center as the reference with which to crop the spine area into several parts to predict coordinates. Zhang et al. [18] were first to obtain a vertebra-bounding box sequence using R-CNN, and then to predict the coordinates in the boxes. Nonetheless, this approach still lacks robustness as the accuracy of the detection outcome is contingent upon the segmentation performance. An erroneous allocation of landmarks to neighboring regions of interest can result in a failure of detection. Certain investigations posit that the assessment of CAs in both the lateral and posteroanterior (PA) perspectives is essential, as the characteristics derived from one view may enhance the informational content obtained from another. To extract multi-view features, the X-modules [21], joint-view network [22], and feature fusion module [23] have been proposed. However, such methods are not applicable in clinical settings where only PA X-rays are available. Moreover, regression-based methods attempt to directly learn the mapping between the input image and coordinates, which increases the difficulties in model optimization during training and affects the accuracy of predicted landmarks.

Unlike regression-based approaches, recent advancements in the field have shown that generating heatmaps of landmarks by deep learning models, rather than outputting landmark coordinates directly, has resulted in greater effectiveness and resilience [24,25]. Zhang et al. [24] proposed SpineHRNet to acquire heatmaps of endplate vertices and end vertebrae. The endplate landmark heatmap includes four channels, and each channel includes a group of 18 landmarks, while the end vertebrae heatmap consists of one channel including all end vertebrae. Their CA results demonstrate a substantial correlation with the ground truth (GT) regardless of variations in image quality or curve patterns. However, two neighbor endplate landmark spots can merge into one spot when they are close to each other, leading to missing landmarks. Meng et al. [25] developed SpineHRNet+, which predicts the heatmap for each landmark to solve this problem. However, the endplate landmark prediction sometime locates the rib area when the vertebra is close to the rib. To ameliorate this problem, they use a spine segmentation model to constrain the location of the output landmarks. Despite the remarkable accuracy exhibited by SpineHRNet+, the procedure is rather complex, encompassing several networks. Consequently, the compounding of errors at every stage can impact the results.

The purpose of the study is to develop a model called SpineHRformer that can accurately estimate the CAs from coronal X-rays, regardless of the image quality or the severity of scoliosis. The working hypothesis is that the SpineHRformer, which consists of three stages, (1) HRNet [27] for multi-scale feature extraction, (2) a transformer encoder [28] for local and global feature extraction, and (3) an output head for the heatmap prediction of landmarks, will be effective in estimating CAs accurately.

## 2. Materials and Methods

### 2.1. Dataset and Image Pre-Processing

We enrolled spinal deformity patients from two territory-wide tertiary scoliosis referral centers (Duchess of Kent Children’s Hospital at Sandy Bay and Queen Mary Hospital in Pok Fu Lam) between December 2019 and November 2020. The study received approval from the local institutional review board (UW15-596), and was conducted in accordance with the Helsinki Declaration of 1975, as revised in 2013. All participants signed their written informed consent. Exclusion criteria included psychological and systematic neural disorders, congenital deformities, prior spinal surgeries, posture- or mobility-impairing trauma, and oncological diseases. Technicians took photos or screenshots of anonymized upper body PA X-rays, ensuring parallel image planes to the screen, and excluded patient demographic information.

The end vertebrae and CAs obtained in clinical routines were used as GT to validate the accuracy of CA predictions. Senior surgeons, having over 20 years of clinical experience, manually marked 2 vertices of each endplate in coronal view from the 7th cervical vertebra (C7) to the 5th lumbar vertebra (L5), i.e., 72 points, to obtain the GT of landmarks. Our self-developed Python-script marking tool was used for landmark placement and coordinate exportation. Inter-rater agreement and consistency between specialists were confirmed by testing 50 images and their labels. From the pool of 2135 recruited participants, we excluded 16 owing to congenital deformities, and a further 185 due to degenerative deformities. A total of 1934 X-ray images (74% female; age range 10–18) were utilized in this study, of which 1550 were allocated for model training and the remaining 384 were used to assess the performance of the model. No patient appeared both in training and testing cohorts.

The CAs were classified according to the position in the spine, i.e., thoracic CA (TCA) or lumbar CA (LCA). The major curve CA (MCA) was determined by the maximum CA observed among the TCA and LCA, acting as a determinant of scoliosis severity. Table 1 delineates the CA thresholds, their corresponding severity classifications and their clinical interventions.

To standardize the images, we automatically resized them to a consistent dimension of 512 pixels in height and 256 pixels in width, containing the entire spine. Throughout the training process, we adopted data augmentation techniques, including: (1) random flipping with a probability of 0.5, (2) scaling within the range of 0.8 to 1.2, (3) rotation within the range of −5° to 5°, and (4) horizontal/vertical translation within the range of −10 to 10 pixels. The endplate landmark and the end vertebrae coordinates were also adjusted accordingly.

### 2.2. SpineHRformer

To ensure the accurate learning of landmark heatmaps, our model is designed and constructed in three stages, as illustrated in Figure 1. The first stage utilizes HRNet for multi-scale feature extraction. A transformer encoder is applied in the subsequent stage to extract local and global features. In the final stage, an output head was incorporated to output heatmaps of end vertebrae or endplate landmarks. The heatmap outputs for endplate landmarks consist of 72 channels, with each channel representing a unique landmark. In contrast, the end vertebrae were depicted through a single channel encompassing all points.

#### 2.2.1. HRNet 

Our SpineHRformer first extracts the multi-scale features using the HRNet [29] (as shown in the red block in Figure 1). It comprises parallel subnetworks operating at varying resolutions, featuring HR-modules that facilitate inter-resolution information exchange across multiple feature maps. The architecture of HRNet adopted in our SpineHRformer contains three stages of HR-modules, each executing multi-scale fusion, thereby merging features at different resolutions and making one more branch. Importantly, this fusion process combines low-resolution semantic representation with high-resolution low-level features, generating relatively robust representations.

#### 2.2.2. Transformer Encoder

After extracting the multi-scale features by HRNet, the transformer encoder is used to extract the local and global features. As shown in the gray block in Figure 1, the transformer encoder consists of several (N = 4) transformer encoder layers. The inputs are the feature maps obtained from the HRNet.

To store the relative position of the features in the sequence, the position embedding (denoted by symbol 
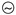
 in Figure 1) is generated by position encoder, which is formed by:(1)$$PE\left(pos,\;2i\right)=sin\left(\frac{pos}{{10,000}^{\frac{2i}{d}}}\right),$$
(2)$$PE\left(pos,\;2i+1\right)=cos\left(\frac{pos}{{10,000}^{\frac{2i}{d}}}\right),$$
where *PE* is the position embedding, and *pos* refers to the position of the feature within the feature map, encompassing the height or width of the feature. *i* is the dimension of position embedding. *d* denotes the maximum dimension of the position embedding, which is equivalent to half the channel number of the feature map. Upon procuring the position embedding, which possesses identical dimensions to the input feature map, both the input feature map and position embedding are reshaped into sequences and subsequently added. The sequence noted as X will be fed into the core of the transformer, i.e., multi-head attention.

In one head self-attention, the sequence $X\in R^{L\times d}$ should be first projected into queries $Q\in R^{L\times d}$, keys $K\in R^{L\times d}$, and values $V\in R^{L\times d}$ by three learnable matrices, $W_{q},W_{k},W_{v}\in R^{d\times d}$. The self-attention output is calculated by:(3)$$Attention\left(Q,K,V\right)=softmax\left(\frac{QK^{T}}{\sqrt{d}}\right)V$$

Expanding to multi-head attention, each attention head generates an attention output. The outputs are concatenated, and a linear layer is utilized to produce an output of the same dimensions as the sequence ***X***. Following the residual connection and layer normalization, a fully connected layer activated by the ReLU activation function is used. After another residual connection and layer normalization, the encoder layer outputs are obtained. The output will be fed into the next encoder layer until the last one and then reshaped to the dimension of the input feature map. 

#### 2.2.3. Output Head

After the HRNet and the transformer encoder, an output head, i.e., a convolution layer, is attached to obtain the final landmarks. For the endplate landmark detection, there are 72 channels in the output, while for the end vertebrae detection, the output contains one channel, as shown in Figure 1.

### 2.3. Performance Evaluation and Statistical Analysis

The performance of the proposed SpineHRformer was evaluated by assessing its capabilities in landmark detection, CA measurement, and severity classification. 

The difference between the predicted and ground truth landmarks was measured using the mean Euclidean distance (MED) of a single landmark and all landmarks, respectively. The MED of the *n^th^* landmark is defined as:(4)$$MED\left(n\right)=\frac{1}{M}\sum_{i=1}^{M}\sqrt{{\left(x_{n,i}-{\hat{x}}_{n,i}\right)}^{2}+{\left(y_{n,i}-{\hat{y}}_{n,i}\right)}^{2}}$$

The MED of all the landmarks is defined as:(5)$$MED=\frac{1}{N\times M}\sum_{n=1}^{N}\sum_{i=1}^{M}\sqrt{{\left(x_{n,i}-{\hat{x}}_{n,i}\right)}^{2}+{\left(y_{n,i}-{\hat{y}}_{n,i}\right)}^{2}}$$where $\left(x_{n,i},y_{n,i}\right)$ and ${(\hat{x}}_{n,i},{\hat{y}}_{n,i})$ denote the GT and predicted coordinate of the $n^{th}$ landmark on the $i^{th}$ image, respectively, N is the number of landmarks in an image, and *M* is the number of images. The $MED\left(n\right)$ and MED provide a straightforward evaluation of landmark detection deviation in comparison to the GT.

The Pearson correlation coefficient (r-value) is used to evaluate the strength of the linear relationship between the CA prediction and the GT. For two positively correlated variables, a higher r-value, approaching 1, signifies a stronger linear positive correlation. Additionally, to assess the validity of the CA prediction, linear regression analysis was performed between the GT and the predictions concerning MCA, TCA, and LCA. The regression line, the 95% confidence interval of the predictions, and the perfect correspondence between the predictions and GT are presented.

Confusion matrix analyses were performed for the 3-level severity classification (normal–mild, moderate, and severe) of both SpineHRformer and SpineHRNet+. The matrix elements represent the proportion of correct and incorrect predictions for each of the three classes, with rows signifying the true class and columns denoting the predicted class. This facilitates the comparison of the model’s performance across distinct classes.

## 3. Experiments and Results

### 3.1. Training

The endplate landmark detection model and the end vertebrae detection model were trained separately. For both training processes, the Adam optimizer [30] was adopted for model optimization. The cosine annealing was used to adjust the learning rate, [31] with the minimum learning rate of 0.00001, and the MSE loss was used. 

### 3.2. Endplate Landmark Detection and CA Results

To validate the performance of the proposed SpineHRformer in landmark detection, we counted the Euclidean distance (pixels) between each predicted landmark and its corresponding annotation in each X-ray. The MED for each landmark was then calculated by averaging over all X-rays. Figure 2 compares the performance of SpineHRformer and SpineHRNet+ [25] using a bar plot. As shown, for most of the landmarks, SpineHRformer outperforms SpineHRNet+ with smaller MEDs. SpineHRformer demonstrates an average distance of 2.47 pixels across all landmarks, with 2.29 pixels in the thoracic region and 2.76 pixels in the lumbar region. In contrast, SpineHRNet+ exhibits an average distance of 2.74 pixels overall, 2.49 pixels in the thoracic region, and 2.91 pixels in the lumbar region.

The performance of MCA, TCA and LCA auto-measurement was analyzed by linear regression and shown in Figure 3, where Figure 3a–c present the regression analysis results of SpineHRformer for MCA, TCA, and LCA auto-measurement, respectively, while Figure 3d–f present the corresponding regression analysis results of SpineHRNet+. In each subfigure of Figure 3, the *x*-axis denotes the predicted CAs and the *y*-axis denotes the GT CAs. The r-value for each measurement was calculated. The analysis reveals that SpineHRformer has higher correlation coefficients (MCA: r = 0.86; TCA: r = 0.84; LCA: r = 0.74) than SpineHRNet+ (MCA: r = 0.83; TCA: r = 0.81; LCA: r = 0.7). At the same time, the regression coefficients and intercepts for SpineHRformer are closer to 1 and 0, respectively, indicating that the results obtained by SpineHRformer are more accurate. Furthermore, the r-value for MCA was found to be the closest to 1 for SpineHRformer, followed by TCA, and finally LCA. Therefore, the results obtained from SpineHRformer are the most accurate for MCA, followed by TCA and LCA.

Following the clinical implications and interventions in [25], no intervention was required for the normal–mild cohort, while bracing may be required for the moderate patients, and surgical intervention may be required for the severe patients (Table 1). We therefore classified the severity into 3 categories, namely normal–mild, moderate, and severe. The confusion matrices are shown in Figure 4 to validate the severity classification performance. In a confusion matrix, the horizontal axis corresponds to the predicted severity outcomes, whereas the vertical axis denotes the clinical results obtained based on X-rays. The principal diagonal components signify the proportion of accurate predictions for each respective category, while the non-diagonal components convey the ratio of misclassified instances in relation to the true quantity of instances for each category. The sensitivities of the normal–mild and severe levels for SpineHRformer (normal–mild = 0.93 and severe = 0.74) outperform SpineHRNet+ (normal–mild = 0.74 and severe = 0.70).

In Figure 5, the ground truth (blue points) and the prediction (red points) of endplate landmarks are shown. The samples include different severity levels as described in Table 1. Compared with SpineHRNet+, SpineHRformer’s predicted points are more concentrated to the spine area, as shown in Figure 5a,e. At the same time, as shown in Figure 5b,c,f,g), the predictions of SpineHRformer are closer to the GT than SpineHRNet+. On low-quality images, such as Figure 5d, SpineHRformer still shows high performance and exceeds SpineHRNet+ in Figure 5h.

## 4. Discussion

In this study, an automatic measurement method for precise CA determination is presented, consisting of endplate landmark detection, end vertebrae detection, and CA calculation. The proposed SpineHRformer is trained separately to achieve endplate landmark and end vertebrae detection. Compared to SpineHRNet+, SpineHRformer exhibits superior performance in predicting accurate endplate landmarks and CAs. Moreover, SpineHRformer achieves higher-sensitivity results in normal–mild and severe cases for severity prediction, indicating its potential clinical applicability. 

In comparison to SpineHRNet+, our proposed SpineHRformer demonstrates superior performance in endplate landmark detection. This is evidenced by a reduced deviation of predicted landmarks near ribs and their increased proximity to the GT. The enhancement of performance can be attributed to the incorporation of a transformer encoder in our model, which differentiates it from SpineHRNet+. In our proposed SpineHRformer, the input to the transformer encoder is derived from the feature maps extracted by HRNet. Each channel is treated as a patch, which is then flattened and transformed into a sequence of embeddings. The self-attention mechanism enables the transformer encoder to capture local features by focusing on the relationships between nearby patches. Since the self-attention mechanism compares each patch with every other patch, it can discover local patterns within neighboring patches and assign higher weights to relevant nearby patches. At the same time, the transformer encoder can capture global features by considering the relationships between all patches in the image [32]. The self-attention mechanism allows the model to incorporate the overall context of the image and assign higher weights to patches that are important for the global context. Therefore, with the transformer encoder, SpineHRformer can better extract features and obtain better endplate landmark prediction results than SpineHRNet.

Both SpineHRformer and SpineHRNet+ exhibit lower measurement accuracy for landmarks in the lumbar region compared to those in the thoracic region. This is due to the larger size of lumbar vertebrae and the increased spacing between them, resulting in a larger area of interest and higher requirements for feature extraction in the models. Furthermore, the contents of the intestine can affect the image sharpness and then influence the model’s performance. Consequently, the measurement accuracy of TCA is better than that of LCA for both SpineHRformer and SpineHRNet+. Owing to the enhanced accuracy of endplate landmark predictions by SpineHRformer in comparison to SpineHRNet+, SpineHRformer demonstrates a superior performance in the automatic detection of CAs.

Regarding severity classification, SpineHRformer surpasses SpineHRNet+ in both the normal–mild and severe categories, while SpineHRNet+ exhibits superior performance in the moderate category. This discrepancy can be ascribed to SpineHRformer’s demand for a more extensive training dataset, as the available X-ray images in our dataset fall short of allowing the model to reach its peak performance [33]. Consequently, the model’s limitation stems from the data size; thus, either an increased volume of training data or further refinement of the model is necessary to enhance its effectiveness on limited datasets.

## 5. Conclusions

In this study, we have devised a novel model called SpineHRformer that can effectively measure CAs. When contrasted with the preceding SpineHRNet+ model, our proposed model demonstrates enhanced accuracy. SpineHRformer presents considerable promise and scholarly merit in aiding medical professionals in the diagnosis of scoliosis.

## Figures and Tables

**Figure 1 bioengineering-10-01333-f001:**
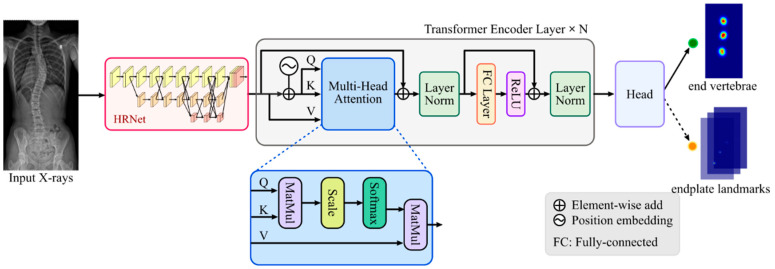
Overview of the architecture of SpineHRformer for end vertebrae and endplate landmark detection. The transformer encoder comprised 4 transformer encoder layers. The Q, K, and V are queries, keys, and values of the self-attention, respectively.

**Figure 2 bioengineering-10-01333-f002:**
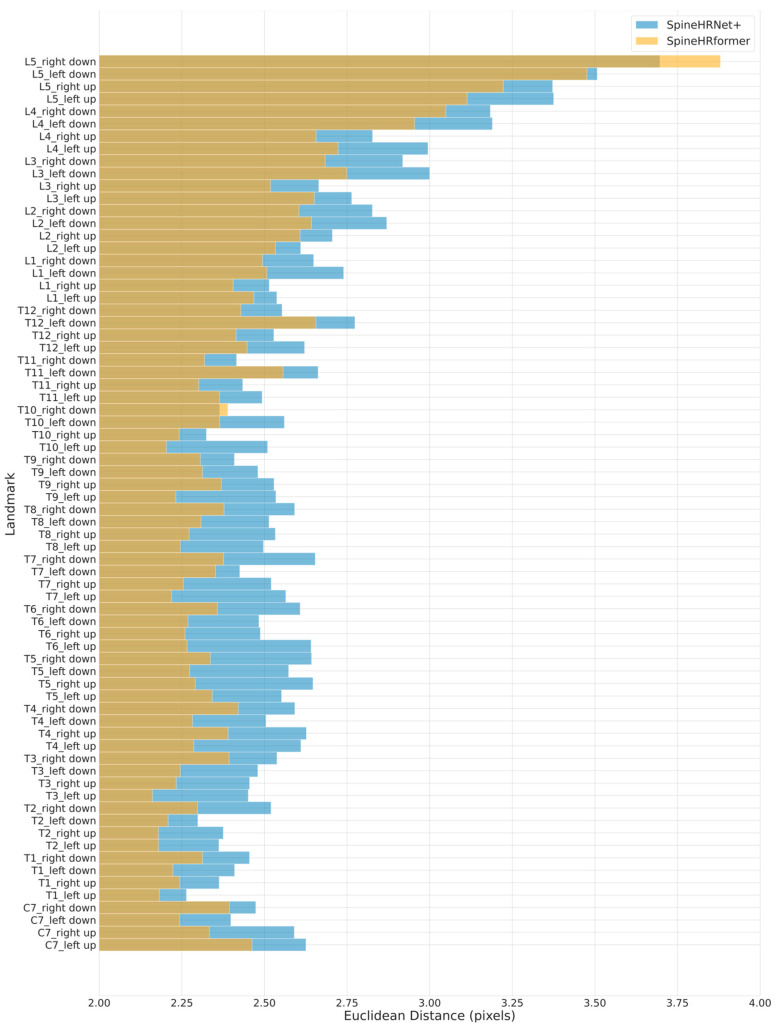
Statistical evaluation of SpineHRformer against SpineHRNet+ on vertebra endplate landmark detection.

**Figure 3 bioengineering-10-01333-f003:**
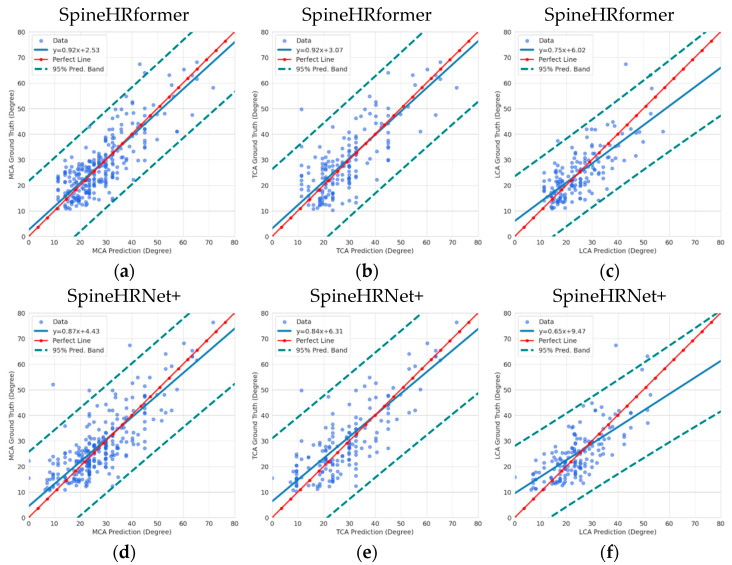
Linear regression analysis of MCA, TCA an LCA. (**a**–**c**) Linear regression analysis of MCA, CAT, and CAL obtained from SpineHRformer. (**d**–**f**) Linear regression analysis of MCA, CAT, and CAL obtained from SpineHRNet+.

**Figure 4 bioengineering-10-01333-f004:**
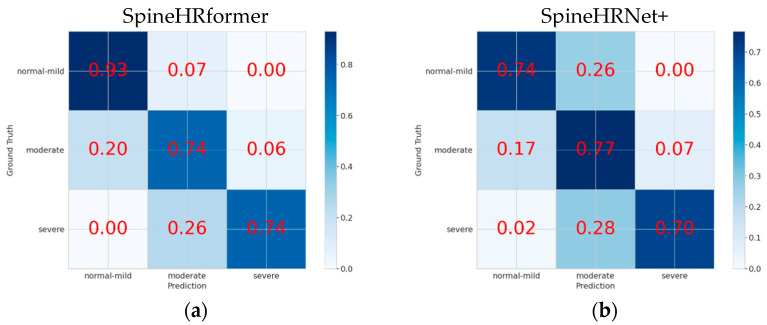
Confusion matrix analyses for severity classification. (**a**) Confusion matrix of SpineHRformer. (**b**) Confusion matrix of SpineHRNet+.

**Figure 5 bioengineering-10-01333-f005:**
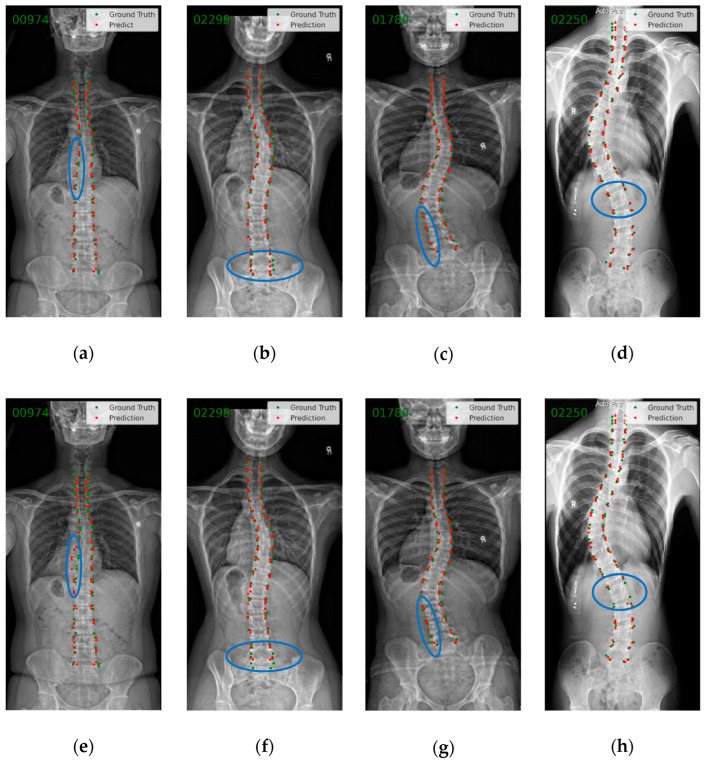
Visual comparison between SpineHRNet+ and the proposed SpineHRformer. The blue ellipses show the difference between the results of the two methods. The letter “R” on each X-ray denotes the right side of the body. (**a**–**d**) Landmark detection results using SpineHRformer on normal, mild, moderate, and severe X-rays, respectively. (**e**–**h**) Corresponding results obtained using SpineHRNet+.

**Table 1 bioengineering-10-01333-t001:** Severity levels associated with CA.

Severity Level	Cobb Angle	Clinical Intervention
Normal-mild	$CA\leq 20^{{}^{\circ}}$	No intervention required.
Moderate	$20^{{}^{\circ}}<CA\leq 40^{{}^{\circ}}$	May require bracing to prevent curve progression.
Severe	$CA>40^{{}^{\circ}}$	Surgical intervention may be required

## Data Availability

The data presented in this study are available on request from the corresponding author. The data are not publicly available due to the ethical requirements of the affiliation.

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
