# Peer review of "SpineHRformer: A Transformer-Based Deep Learning Model for Automatic Spine Deformity Assessment with Prospective Validation"

_bioengineering, 2023, doi:10.3390/bioengineering10111333_

Round 1
Reviewer 1 Report
Comments and Suggestions for Authors
Title. I think it is better that they indicate the type of clinical trial carried out.
In the introduction, line 99, do not say the contribution but rather the purpose of the study and the working hypothesis.
Regarding the material and methods, say how many rX were used for the comparison.
Discussion
Line 312- Indicate the limitations of your study.
Add a conclusions section after lines 315
Reviewer 2 Report
Comments and Suggestions for Authors
My comments are in the file.

My comments are in the file.
Reviewer 3 Report
Comments and Suggestions for Authors
The authors describe an interesting and helpful line of research.
I have some comments:
Introduction:
Briefly define the term "deep learning" as you are using it.
Can you state a specific hypothesis that your study was designed to prove?
Material & Methods:
No specific details regarding patient consent. Your manuscript does not contain a complete IRB statement regarding ethics board approval. Original articles need to contain a statement about the Helsinki Declaration of 1975, as in the example given here: “This study was approved by the human subject’s ethics board of XXXXX and was conducted in accordance with the Helsinki Declaration of 1975, as revised in 2013.
The methodology in this study has certain limitations due to the lack of description and citation of the software used in the procedure. While the use of end vertebrae and clinical assessments for validation is noted, it is essential to provide details about the software or tools employed for these manual annotations. This lack of information can lead to questions about the reproducibility and standardization of the landmark identification process. Additionally, it would be beneficial to mention if there was any inter-rater reliability assessment among senior surgeons who manually marked the points to ensure consistency in landmark identification.
Discussion:
The Limitations section should be included stating the weaknesses of the study, the implications for clinical practice and research, and the conclusion,
Round 2
Reviewer 3 Report
Comments and Suggestions for Authors
The article was carefully revised and now it is in accordance for approval.